# MambaTalk: Efficient Holistic Gesture Synthesis with Selective State Space Models

**Zunnan Xu⋆, Yukang Lin∗, Haonan Han∗, Sicheng Yang, Ronghui Li, Yachao Zhang†, Xiu Li†**
Shenzhen International Graduate School, Tsinghua University
University Town of Shenzhen, Nanshan District, Shenzhen, Guangdong, P.R. China

## Abstract

Gesture synthesis is a vital realm of human-computer interaction, with wide-ranging applications across various fields like film, robotics, and virtual reality. Recent advancements have utilized the diffusion model to improve gesture synthesis. However, the high computational complexity of these techniques limits the application in reality. In this study, we explore the potential of state space models (SSMs). Direct application of SSMs in gesture synthesis encounters difficulties, which stem primarily from the diverse movement dynamics of various body parts. The generated gestures may also exhibit unnatural jittering issues. To address these, we implement a two-stage modeling strategy with discrete motion priors to enhance the quality of gestures. Built upon the selective scan mechanism, we introduce MambaTalk, which integrates hybrid fusion modules, local and global scans to refine latent space representations. Subjective and objective experiments demonstrate that our method surpasses the performance of state-of-the-art models. Our project is publicly available at `https://kkakkkka.github.io/MambaTalk/`.

## 1 Introduction

Gesture synthesis is a critical area of research in human-computer interaction (HCI), which has very broad application prospects, such as film, robotics, virtual reality, and digital human development [24]. The task is challenging due to the variable correlation between speech and gestures, as the same spoken content can elicit markedly different gestures among speakers. Meanwhile, the generated gestures should synchronize with the speaker's rhythm, emotional cues, and intentions [31, 1, 8, 42].

Recent works in co-speech gesture generation have shown great progress [11, 46, 67, 66, 2, 62]. By introducing new datasets [71] and more modalities [70, 33], previous work achieved end-to-end gesture generation based on RNN-based models [11, 46]. With the success of transformer in nature language processing [58] and video sequence modeling [26, 27, 75], recent works [8, 43, 55] leverage the power of attention mechanism to generate more expressive gestures that better synchronize with speech. By further combining emotional and style related features, EMoG [69] achieve better quality gesture generation. With the development in human recognition model [37], EMAGE [32] proposes a masked audio-gesture modeling strategy to enhance unified holistic gesture synthesis. Recently, with the development of diffusion model in generative tasks [39, 20, 40], the latest works [74, 2, 62, 7, 65] have applied the diffusion model to gesture synthesis, significantly improving the diversity of generated gesture. DiffuseStyleGesture [63] presents a diffusion model-based approach for generating diverse co-speech gestures by incorporating cross-local attention and self-attention mechanisms, and utilizing classifier-free guidance for style control. DiffuseStyleGesture+ [66] further considers the text modality as an additional input and utilizes channel concatenation to merge the text feature

---

⋆Work done as intern at Tecent.
∗Equal Contribution.
†Corresponding author.

38th Conference on Neural Information Processing Systems (NeurIPS 2024).

with the audio feature. Deichler *et al.* [9] also incorporates the text modality as an additional input and employs contrastive learning to enhance the features. However, the exploration of generation for co-speech gesture sequences with low latency remains relatively uncharted, constraining its application in dynamic, interactive environments. RNN-based models often struggle with the long-term forgetting issue [54, 29], which impairs their ability to generate long sequences of gestures effectively. Additionally, these models may produce gestures that lack variability, tending towards an average representation [36]. Transformer-based models depend heavily on subtle positional encoding to capture the order of input elements [44, 50, 73]. Meanwhile, their computational complexity, which grows quadratically with the length of the input sequence, poses a challenge for generating long sequences of gestures. For the diffusion-based model, the intricate sampling strategy and iterative process lead to high computational expenses [48], which hinder their broad adoption in gesture generation scenarios.

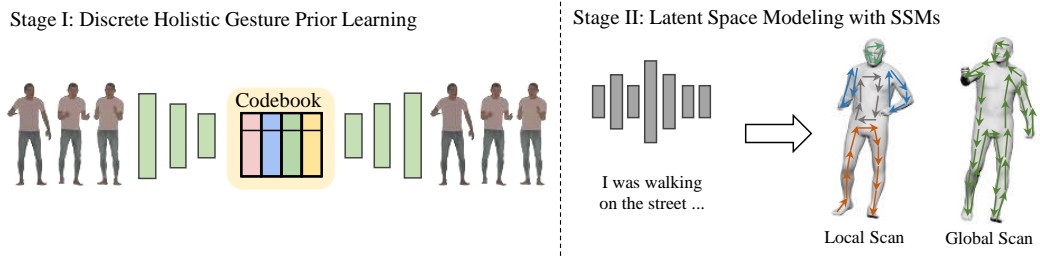

Figure 1: Our two-stage method for co-speech gesture generation with selective state space models. In the first stage, we construct discrete motion spaces to learn specific motion codes. In the second stage, we develop a speech-driven model of the latent space using selective scanning mechanisms.

State space models (SSMs) have recently shown significant potential in addressing challenges related to modeling sequences with low latency [12]. Inspired by continuous state space models from control systems and enhanced by HiPPO initialization [13], SSMs [16] show promise in addressing long-term forgetting issue. These advancements have been integrated into large-scale representation models [38, 41]. Some pioneering works have applied SSMs for tasks like language understanding [38, 41], content-based reasoning [12], and visual recognition [35, 57]. In our work, we further explore the potential of SSMs in co-speech gesture synthesis. We observe that directly applying the selective scan mechanism from Mamba [12] to gesture generation as a sequence modeling model would result in jittery outputs. To refine the generated gestures, we propose a two-stage modeling strategy. In the first training stage, we enhance the discrete motion priors derived from VQVAEs [53] by integrating velocity and acceleration losses. In the second stage, by utilizing motion priors from VQVAEs, we introduce individual learnable queries for different body parts, thereby alleviating the jittering issue. Meanwhile, considering that the direct application of Mamba encounters the challenge of limb movements across different body parts tending to average out, we propose a hybrid scanning approach in the second stage to enhance the motion representation in the latent space. Specifically, we refine the design of spatial and temporal modeling within latent spaces by introducing a global-to-local modeling strategy and integrating attention mechanisms along with a selective scanning approach into the framework's design. Considering the significant differences in deformation and movement patterns among different body parts [32], we propose local and global scan modules for refining the latent space representations of the movements across various body parts. These approaches enable dynamic interaction and iterative refinement of different body parts while maintaining low latency, leading to more diverse and rhythmic gestures. Our contributions can be summarized as below:

- We are the first to explore the potential of the selective scan mechanism for co-speech gesture synthesis, achieving a diverse and realistic range of facial and gesture animations.
- We introduce MambaTalk, an innovative framework that integrates hybrid scanning modules (e.g., local and global scan). The integration enhances the latent space representations for gesture synthesis, thereby refining the distinct movement patterns across various body parts.
- Extensive experiments and analyses demonstrate the effectiveness of our proposed method.

## 2 Related Work

### 2.1 Co-speech Gesture Generation

Co-speech gesture generation aims to automatically generate gestures based on speech input. Existing approaches can be broadly categorized into three groups: (i) Rule-based methods: These methods rely on pre-defined rules and gesture libraries to generate gestures based on speech features [23, 56]. While offering interpretable results, they require significant manual effort in creating gesture datasets and defining rules. (ii) Statistical models: These approaches leverage data-driven techniques to learn mapping rules between speech and gestures, often employing pre-defined gesture units [22, 25]. While overcoming the limitations of manual rule creation, these methods still rely on handcrafted features. (iii) Deep learning methods: Recent advancements in deep learning have enabled neural networks to capture the complex relationship between speech and gestures directly from raw multimodal data [70, 33, 68, 32]. This progress has established deep learning approaches, particularly recurrent neural networks (RNNs) [70, 33, 61], transformers [4, 45], and diffusion models [2, 74, 51, 72, 21, 6], as the prevailing paradigm for co-speech gesture generation. However, each of these models suffers from certain limitations that hinder their performance. RNNs inherently process sequences in a serial manner, where each timestep's computation depends on the output of the previous timestep. This limits their ability to efficiently handle long sequences and introduces cumulative latency. Meanwhile, RNNs lack inherent parallelism, further restricting their potential for high-speed computation. Transformers consider all positions within a sequence at every timestep, resulting in high computational complexity, especially for long sequences. While diffusion models significantly enhance the diversity of generated outputs, the sampling process is computationally expensive. To overcome these limitations, our method investigates the capacity of selective state space models in the field of gesture synthesis. To the best of our knowledge, we are the first to apply selective state space models to the task of gesture generation.

### 2.2 Selective State Space Models

State Space Models (SSMs) are a novel class of models recently integrated into deep learning for state space transformation [15, 10]. As foundational models evolve, various subquadratic-time architectures have emerged, including linear attention, gated convolution, recurrent models, and structured state space models (SSMs), aimed at mitigating the computational inefficiencies of Transformers when dealing with lengthy sequences. However, these advancements have yet to match the performance of attention mechanisms in critical modalities like language processing.

SSMs draw inspiration from continuous state space models in control systems and, when combined with HiPPO initialization [13], as seen in LSSL [16], show promise in tackling long-range dependency issues. However, the computational and memory demands of the state representation render LSSL impractical for real-world use. To address this, S4 [15] suggests normalizing the parameters into a diagonal structure. This has led to the emergence of various structured SSMs with diverse configurations, such as complex-diagonal structures [17, 14], multiple-input multiple-output (MIMO) support [49], diagonal-plus-low-rank decomposition [19], and selection mechanisms [12]. These models have been incorporated into large-scale representation models [38, 41].

These models primarily focus on the application of SSMs to long-range and sequential data like language and speech, for tasks such as language understanding [38, 41], content-based reasoning [12], and pixel-level 1-D image classification [15]. Recently, some pioneering work [35, 57, 59] have explored their application in visual recognition. We further demonstrate that by incorporating the selective scan mechanism from mamba [12] and the discrete motion priors from VQVAEs [53], our proposed MambaTalk is capable of matching the performance of existing popular holistic gesture synthesis models, highlighting the potential of MambaTalk as a powerful gesture synthesis model.

## 3 Method

We aim to synthesize sequential 3D co-speech gestures from speech signals (e.g., audio and text) using selective state space models. However, simply applying such a model to gesture synthesis leads to severe gesture jittering issues. We also found that maintaining performance is challenging due to the significant variations in movement patterns exhibited by different body parts. To overcome these challenges, we suggest modeling the gesture space using the acquired discrete motion patterns.

Subsequently, we propose to develop speech-conditioned selective state space models within this framework. This approach is designed to enhance the model's robustness against uncertainties that arise from cross-modal discrepancies. As shown in Figure 2, our framework consists of two stages: (i) modeling the discrete gestures and facial motion spaces (§3.2) and (ii) learning speech-conditioned selective state space models (§3.3) to generate 3D co-speech gestures.

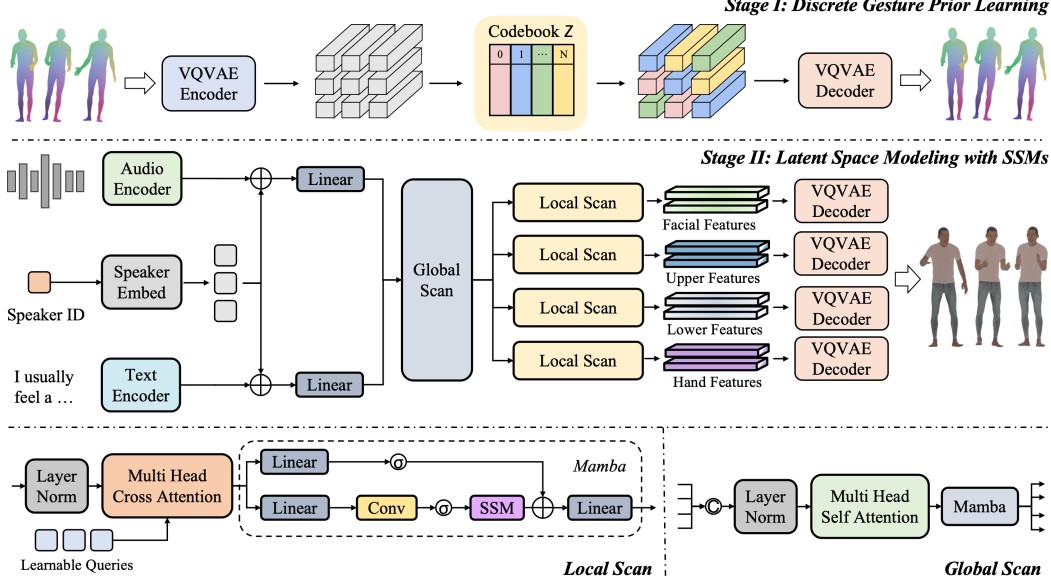

Figure 2: We propose a two-stage method for co-speech gesture generation. We first train multiple VQ-VAEs for face and different parts of body reconstruction. This step learns discrete motion priors through multiple codebooks. In the second stage, we train a speech-driven gesture generation model in the latent motion space with local and global scan modules.

## 3.1 Preliminaries

**Selective State Spaces Model.** In our approach, we adopt the Selective State Spaces model (Mamba [12]) that incorporates a selection mechanism and a scan module (S6). This model is designed to make sequence modeling, as it dynamically selects salient input segments for prediction, thereby enhancing its focus on pertinent information and improving overall performance. Unlike the traditional S4 model, which uses time-invariant matrices $A$, $B$, $C$, and scalar $\Delta$, Mamba introduces selection mechanism that allows for the learning of these parameters from the input data using fully-connected layers. This adaptability enables model to better generalize and perform complex modeling tasks. Mamba operates by defining the state space with structured matrices that introduce specific constraints on the parameters, facilitating efficient computation and data storage. For each batch and each dimension, the model processes the input $x_t$, hidden state $h_t$, and output $y_t$ at each time step $t$. We have $h_0 = \bar{B}_0 x_0$ when $t = 0$. When t > 0, the model's formulation is as follows:

$$
\begin{aligned}
h_t &= \bar{A}_t h_{t-1} + \bar{B}_t x_t, \\
y_t &= C_t h_t,
\end{aligned}
\tag{1}
$$

where $\bar{A}_t$, $\bar{B}_t$, and $C_t$ are matrices and vectors that are updated at each time step, allowing the model to adapt to the temporal dynamics of the input sequence. With discretization, let $\Delta$ denote the sampling interval, $\exp(\Delta A)$ denote the matrix exponential, the transformation of the system's state over one time step can be represented as follows:

$$
\begin{aligned}
\bar{A} &= \exp(\Delta A), \\
\bar{B} &= (\Delta A)^{-1}(\exp(\Delta A) - I) \cdot \Delta B, \\
h_t &= \bar{A} h_{t-1} + \bar{B} x_t,
\end{aligned}
\tag{2}
$$

where $(\Delta A)^{-1}$ denotes the inverse of matrix $\Delta A$, $I$ denotes the identity matrix. The scan module within Mamba is designed to capture temporal patterns and dependencies across multiple time steps by applying a set of trainable parameters or operations to each segment of the input sequence. In our framework, Mamba serves as a sequence modeling tool for decoding gesture actions across different parts of the body. By modifying the decoder's input and the range of features, we utilize Mamba to separately model the global motion features and local motion features of different body parts. These operations are learned during training and assist the model in processing sequential data.

## 3.2 Discrete Gestures and Facial Motion Spaces

To ensure visual realism in motion animations from speech signals, we learn extra motion priors to depict accurate movements and natural expressions. Building on this concept, we propose a method to represent the gesture motion space using multiple discrete codebooks.

**Motion Quantization.** Considering the substantial variations in deformation magnitude and periodicity among various body parts, our approach involves learning multiple codebooks tailored for the reconstruction of distinct body parts. For illustrative purposes, we detail the formulation of a single codebook. Denotes $C$ as the dimensionality of each latent vector, $N$ as the number of vectors in the codebook, for the codebook $\mathcal{Z} = \left\{ \mathbf{z}_k \in \mathbb{R}^C \right\}_{k=1}^N$, we employ a set of allocated items $\{\mathbf{z}_k\}_{k \in \mathcal{S}}$ to represent the holistic gesture motion $\mathbf{M}_t$. Here, $\mathcal{S}$ represents the chosen index sets. The element-wise quantization function $Q(\cdot)$ maps each item $\hat{\mathbf{z}}_t$ in $\hat{\mathbf{Z}}$ to its closest match $\mathbf{z}_k$ in the codebook $\mathcal{Z}$:

$$\mathbf{Z_q} = Q(\hat{\mathbf{Z}}) := \arg\min_{\mathbf{z}_k \in \mathcal{Z}} \|\hat{\mathbf{z}}_t - \mathbf{z}_k\|_2 , \tag{3}$$

where the codebook entries act as the foundational motion elements within the discrete motion space. To establish this, we follow [64, 32] to pre-train a CNN-based Vector Quantized-Variational Autoencoder (VQ-VAE), which comprises an encoder $E$, a decoder $D$, and a context-rich codebook $\mathcal{Z}$. This is done through the self-reconstruction of gesture motions.

The sequence of motions $\mathbf{M}_{1:T}$ is initially transformed into a temporal feature representation $\hat{Z} = E(\mathbf{M}_{1:T}) \in R^{T' \times H \times C}$, where $H$ represents the count of gesture components, $T'$ indicates the quantity of temporal units encoded (with $P = \frac{T}{T'}$ frames per unit). Subsequently, we derive the quantized motion sequence $Z_q \in R^{T' \times H \times C}$ by quantization function $Q(\cdot)$. This function $Q$ maps each element in $\hat{Z}$ to its closest corresponding entry within the codebook $\mathcal{Z}$:

$$\mathbf{Z_q} = Q\left(E\left(\mathbf{M}_{1:T}\right)\right), \hat{\mathbf{M}}_{1:T} = D\left(\mathbf{Z_q}\right). \tag{4}$$

**Training objectives.** For the training of the quantized autoencoder, we employ motion-level losses to mitigate the jittering issue of generated gestures, along with two intermediate losses at the code level:

$$\begin{aligned} \mathcal{L}_{\text{VQ}} = & \mathcal{L}_{rec}(\mathbf{M}, \hat{\mathbf{M}}) + \mathcal{L}_{vel}(\mathbf{M}', \hat{\mathbf{M}}') + \mathcal{L}_{acc}(\mathbf{M}'', \hat{\mathbf{M}}'') \\ & + \left\| \text{sg}(\hat{\mathbf{Z}}) - \mathbf{Z_q} \right\|_2^2 + \left\| \hat{\mathbf{Z}} - \text{sg}\left(\mathbf{Z_q}\right) \right\|_2^2 , \end{aligned} \tag{5}$$

where $\mathbf{M}'$ and $\mathbf{M}''$ means the velocity and acceleration of motion, $sg(\cdot)$ denotes a stop-gradient operation, $\mathcal{L}_{\text{rec}}$ are Geodesic [52] loss and the last two terms are designed to refine the codebook entries. For facial motions, we utilize MSE loss for both velocity ($\mathcal{L}_{\text{vel}}$) and acceleration ($\mathcal{L}_{\text{acc}}$) loss. For body motions, we use L1 loss as $\mathcal{L}_{\text{vel}}$ and $\mathcal{L}_{\text{acc}}$. Additionally, for the foot contact loss, we employ MSE loss as the loss function. These terms work by minimizing the distance between the codebook $Z$ and the embedded features $\hat{Z}$. Given that the quantization function (Equation 3) is non-differentiable, we utilize the straight-through gradient estimator [53] to propagate the gradients.

## 3.3 Speech-Driven Selective State Spaces Gesture Synthesis Model

**Overall Framework.** Utilizing the acquired discrete motion prior, we establish a cross-modal mapping from speech inputs to target motion codes, enabling the generation of realistic co-speech gesture motions. In our approach to speech-driven gesture synthesis, we utilize audio sequences $A = \{a_1, \ldots, a_N\}$ and text sequences $T = \{t_1, \ldots, t_N\}$ as inputs to guide the generation of co-speech gestures $G = \{g_1, \ldots, g_N\}$. Here, $N$ signifies the total frame count, and $g_i \in R^{55 \times 6 + 100 + 4 + 3}$

denotes 55 pose joints in Rot6D, $R^{100}$ FLAME parameters, $R^4$ foot contact labels, $R^3$ global translations for the $i$-th frame. The gesture synthesis model, comprising audio encoders $E_A$ and text encoders $E_T$ and multiple selective state space models $D_B$ for different parts of the body, is trained on the discrete motion space, conditioned on the speech, as shown in Figure 2.

**Speech Feature Extraction.** For audio feature extraction, two CNN-based audio feature extraction networks are employed to respectively extract features from amplitude, raw audio and decoded tokens from Wav2vec2CTC [3]. Considering that the movements of body parts are not closely linked to raw audio, the audio encoder for body parts does not utilize raw audio as input. Specifically, we integrate these features along the channel dimension to obtain audio features $f_A = \{fa_1, ..., fa_N\}$. For processing speech input words, we employ pre-trained FastText [5] to obtain word embeddings, which are then refined by linear projections to produce text features $f_T = \{ft_1, ..., ft_N\}$. We further fuses features from the input modalities (e.g., audio and text features). The speaker ID embeddings $s_{id}$ are first combined with audio and text features through additive operation. By concatenating feature vectors along the channel dimension, we further apply linear transformations to determine the weight factors, and then integrating the features through an element-wise summation. The process can be formalized as:

$$
\begin{aligned}
w_T &= \sigma(W_T \cdot [f_A + s_{id} \cdot \mathbf{1}, f_T + s_{id} \cdot \mathbf{1}]), \\
w_A &= \sigma(W_A \cdot [f_A + s_{id} \cdot \mathbf{1}, f_T + s_{id} \cdot \mathbf{1}]), \\
\bar{f}_T &= w_T \odot f_A + (1 - w_T) \odot f_T, \\
\bar{f}_A &= w_A \odot f_A + (1 - w_A) \odot f_T,
\end{aligned}
\tag{6}
$$

where $\sigma$ denotes the softmax operation, $\odot$ denotes the Hadamard product, and $W_T$ and $W_A$ represent linear mapping matrices used to adjust the dimensions of merged features. $\bar{f}_T$ and $\bar{f}_A$ denote the fused features.

**Global and Local Scans.** Recognizing the diverse deformations and motion patterns in various body parts, we propose using global scan module and multiple local scan modules to model the movements of different body parts (e.g., face, hand, upper and lower body) with fused multi-modal features from previous modules. By acquiring the speech features from audio and text encoders, we first improve the perception of motion patterns among them using a global scan module by combining the speech features along the sequence dimension. Subsequently, by utilizing self-attention mechanism($\mathcal{F}_{\text{MHSA}}$), we model the global information across different sequence tokens. Following previous work [32], we establish a set of learnable parameters ($Q_{global}$) and employ masked motion to facilitate the acquisition of global information. Then, we employ self-attention mechanisms to enhance the global information and obtain the refined features. These refined features are fed into Mamba to extract temporal perceptual information. Considering the differences in representation between the body and face, we employ two independent MAMBA models to model the temporal features of the face and body, respectively. We then merge these features using a linear layer to obtain global features. The process can be fomulized as below:

$$
\begin{aligned}
\bar{f}_{global} &= \mathcal{F}_{\text{MHSA}}(Q_{global}), \\
f_{speech} &= \text{Mamba}([\bar{f}_T, \bar{f}_A]), \\
\hat{f}_{global} &= \text{Mamba}(\bar{f}_{global}), \\
f_{global} &= \text{Linear}([f_{speech}, \hat{f}_{global}]),
\end{aligned}
\tag{7}
$$

where [] denotes the concatenation operation of features in the dimension of the sequence. To enhance the generalization of the model, we incorporate the learnable queries to foster the queries' ability to learn motion patterns. As shown in Figure 2, the queries from global scan are integrated with input speech features through a multihead cross-attention mechanism ($\mathcal{F}_{\text{MHCA}}$). This allows queries to learn the most relevant information from the speech input. The process can be formally defined as belows:

$$
f_{\text{refine}} = \mathcal{F}_{\text{MHCA}}(\bar{f}_{global}, [\bar{f}_T, \bar{f}_A]),
\tag{8}
$$

where $f_{\text{refine}}$ denotes the feature of refined learnable queries. Utilizing the extracted perceptual features from various body parts, we proceed to employ Mamba to extract temporal features from the sequence, which can be formalized as belows:

$$
F_{face} = \text{Mamba}(f_{refine}),
\tag{9}
$$

where $F_{\text{face}}$ corresponds to the temporal features of facial motion. The same approach is utilized to generate temporal features for the hand, upper body, and lower body by inputting their respective

perceptual features $f_{hand}$, $f_{upper}$, and $f_{lower}$ into the corresponding Mamba modules. One distinction is that we incorporate an additional self-attention layer before the Mamba layer to enhance the perception of body movements. The local latent features are then fed into their respective VQ-Decoders to produce the final motion predictions.

**Training Objectives.** The model's training objectives are composed of a composite loss function that harmonizes reconstruction and cross-entropy losses. This design aims to augment the accuracy of motion generation, encompassing the face, hands, upper, and lower body. The loss of latent reconstruction, represented by $L_{reclatent}$, is quantified using the Mean Squared Loss (MSELoss). Here, $z_i$ corresponds to the true latent vectors, while $\hat{z}_i$ are the vectors reconstructed by the model. The latent reconstruction loss is expressed as:

$$L_{reclatent} = \frac{1}{N} \sum_{i=1}^{N} \|z_i - \hat{z}_i\|^2, \tag{10}$$

where $N$ denotes the number of frames. Concurrently, to encourage diversity in the generated motions, we optimize the cross-entropy loss for latent code class classification $L_{cls}$. Specifically, we employ Negative Log-Likelihood Loss (NLLLoss), where $y_i$ represents the true class labels for each sample, and $\hat{y}_i$ denotes the model's predicted class labels. This loss is calculated as the negative sum of the logarithm of the predicted probabilities for the correct classes:

$$L_{cls} = -\frac{1}{N} \sum_{i=1}^{N} \sum_{c=1}^{C} y_{ic} \log(\hat{y}_{ic}), \tag{11}$$

where $N$ signifies the total number of frames, $C$ is the total number of classes, and $y_{ic}$ is a binary indicator of whether class $c$ is the correct label for sample $i$. The total loss $L$ is a weighted sum of the categorical and latent reconstruction losses, with $\alpha$ and $\beta$ serving as balance hyper-parameters:

$$L = \alpha L_{cls} + \beta L_{reclatent}, \tag{12}$$

where $\alpha = 1$ and $\beta = 3$ for hands, upper and lower body motion. For facial motion, we set $\alpha = 0$ and $\beta = 3$. By optimizing the total loss, the model is trained to generate diverse gesture results.

## 4 Experiments

### 4.1 Experiments Setting

We train and evaluate on the BEAT2 dataset proposed by [32]. BEAT2 contains 60 hours of data with high finger quality for 25 speakers (12 female and 13 male). The dataset comprises 1762 sequences, each with an average duration of 65.66 seconds. Each sequence includes a response to a daily inquiry. We split datasets into 85%/7.5%/7.5% for the train/val/test set. We follow previous work [32] to select data from Speaker 2 for training and validation to ensure fair comparison.

### 4.2 Implementation Details

We utilize the Adam optimizer with a learning rate of $2.5 \times 10^{-4}$. To maintain stability, we apply gradient norm clipping at a value of 0.99. In the construction of the VQVAEs, we employ a uniform initialization for the codebook, setting the codebook entries to feature lengths of 512 and establishing the codebook size at 256. The numerical distribution range for the codebook initialization is defined as $[-1/\text{codebook\_size}, 1/\text{codebook\_size})$. The codebook is solely updated during the first stage, and in the second stage of training for the speech-to-gesture mapping, the codebook remains frozen. The VQVAEs are trained for 200 epochs, with a learning rate of $2.5 \times 10^{-4}$ for the first 195 epochs, which is then reduced to $2.5 \times 10^{-5}$ for the final 5 epochs. During the second stage, the model is trained for 100 epochs. All experiments are conducted using one NVIDIA A100 GPU.

### 4.3 Metrics

To evaluate the realism of body gestures, we employ Fréchet Gesture Distance (FGD) [70] to measure the proximity of the distribution between the ground truth and generated gestures. Subsequently, Diversity [28] is quantified by computing the average L1 distance across multiple gesture clips. The

synchronization between speech and motion is achieved using Beat Constancy (BC) [30]. For facial motions, we assess positional accuracy by calculating the vertex Mean Squared Error (MSE) [60]. Additionally, the difference between the ground truth and the generated facial vertices is measured using the vertex L1 difference (LVD) [68]. *More details about metrics and efficiency analysis are provided in the supplementary materials.*

## 4.4 Quantitative Results

As shown in Table 1, our method attains the lowest FGD and highest BC when compared to the previously top-performing method. This highlights the superior ability of MambaTalk in discerning and correlating the audio-motion beats. The lowest FGD also emphasizes the high quality and naturalness of our generated movements, showing the ability of MambaTalk to capture real motion dynamics. This also demonstrates the authenticity of our generated motions, affirming the successful capture of inherent motion characteristics. Some results are marked as "-" because these methods can not generate facial movements. Moreover, our method outperforms previous methods in terms of MSE and LVD, with substantial improvements of 18.11% and 8.72%, respectively. These two enhancements highlight the superior accuracy and fidelity of our method in capturing fine-grained details, affirming its efficacy in synthesizing realistic and authentic facial motions.

Table 1: Quantitative results on BEAT2. FGD (Frechet Gesture Distance) multiplied by $10^{-1}$, BC (Beat Constancy) multiplied by $10^{-1}$, Diversity, MSE (Mean Squared Error) multiplied by $10^{-7}$, and LVD (Learned Vector Distance) multiplied by $10^{-5}$. The best results are in bold.

| Methods | Venue | FGD ↓ | BC ↑ | Diversity ↑ | MSE ↓ | LVD ↓ |
|---|---|---|---|---|---|---|
| Non-facial Gesture Synthesis | | | | | | |
| S2G [11] | ICRA 2019 | 28.15 | 4.683 | 5.971 | - | - |
| Trimodal [70] | TOG 2020 | 12.41 | 5.933 | 7.724 | - | - |
| HA2G [34] | CVPR 2022 | 12.32 | 6.779 | 8.626 | - | - |
| DisCo [31] | ACMMM 2022 | 9.417 | 6.439 | 9.912 | - | - |
| CaMN [33] | ECCV 2022 | 6.644 | 6.769 | 10.86 | - | - |
| DiffStyleGesture [63] | IJCAI 2023 | 8.811 | 7.241 | 11.49 | - | - |
| Holistic Gesture Synthesis | | | | | | |
| Habible *et al.* [18] | IVA 2021 | 9.040 | 7.716 | 8.21 | 8.614 | 8.043 |
| TalkShow [68] | CVPR 2023 | 6.209 | 6.947 | **13.47** | 7.791 | 7.771 |
| EMAGE [32] | CVPR 2024 | 5.512 | 7.724 | 13.06 | 7.680 | 7.556 |
| MambaTalk (Ours) | - | **5.366** | **7.812** | 13.05 | **6.289** | **6.897** |

## 4.5 Qualitative Analysis

**User Study.** We conduct a user study to assess the visual quality of the generated co-speech 3D gestures. For each method under comparison, we produce 10 gesture samples, which were then converted into video clips for evaluation by 39 participants. In each evaluation session, participants are presented with 20 seconds video clips generated by various models. They are instructed to assess the clips across the following dimensions: (i) naturalness, (ii) appropriateness, (iii) synchrony and (iv) smoothness. For naturalness, they evaluate the similarity of the generated gestures to those made by humans, paying attention to the authenticity and smoothness of the movements. In terms of appropriateness, they consider the alignment of the gestures with the spoken content, taking into account both the explicit meaning and the underlying semantics. For synchrony assessment, they examine the timing of the gestures in relation to the speech rhythm, audio, and facial expressions to ensure a harmonious and integrated performance. For smoothness, they assess the gestures for any abrupt stops or unnatural jerks that might indicate a lack of fluidity in motion. We mainly compare two state-of-art methods with our proposed method (with and without VQVAE): CaMN [33], EMAGE [32], and the ground truth. As presented in Table 2, our method's average scores are higher than previous methods.

**Visualization.** As depicted in Figure 3, our approach yields gestures that exhibit enhanced rhythmic alignment and a more natural appearance. For instance, when conveying "we were", our method

Table 2: User study results on naturalness (human likeness), appropriateness (the degree of consistency with the speech content), synchrony (the level of synchronization with the speech rhythm) and smoothness (the fluency of actions). The rating score range is 1-5, with 5 being the best. "Avg." denotes the average scores. ↑ indicates the higher the better.

| Methods | Naturalness↑ | Appropriateness↑ | Synchrony↑ | Smoothness↑ | Avg. |
|---|---|---|---|---|---|
| CaMN [33] | 3.08 | 3.34 | 3.25 | 3.50 | 3.29 |
| EMAGE [32] | 3.85 | 4.04 | 3.89 | 4.21 | 3.99 |
| **Ours**w/o VQVAE | 1.24 | 1.24 | 1.18 | 1.29 | 1.24 |
| **Ours** | 4.04 | 4.00 | 3.91 | 4.35 | 4.08 |
| Ground Truth | 4.57 | 4.54 | 4.23 | 4.63 | 4.50 |

instructs the subject to hold both hands in front of the chest, a nuanced detail absent in both CaMN and EMAGE's outcomes, where either one or both arms hang down. Additionally, when representing "no place to", our method aligns with the ground truth by extending both arms upwards, whereas CaMN and EMAGE have their arms tucked in next to the body. In the case of "up", our generated result raises the right arm in alignment with the semantics of movement. In the context of "moving around" where our left and right arm swings may differ from the ground truth, the overall movement remains consistent.

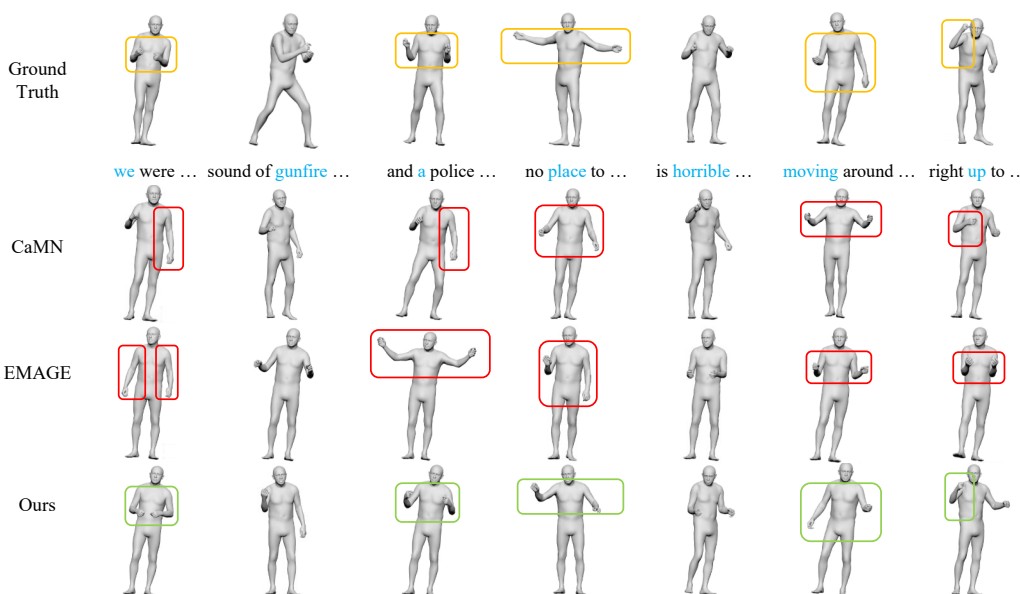

Figure 3: Visualization of the gestures generated by CaMN, EMAGE and our method. Unreasonable results are indicated by red boxes and reasonable ones by green boxes.

Interestingly, for "sound of gunfire", a difficult semantic for the model to learn, our method still generates the result of the character's right hand clenched in a fist and the arm bent to indicate a tense situation. For the emotion of fear expressed by "is horrible", the result of our method is similar to the ground truth, with the character's hands hanging down and face facing downward, which is a visual representation of the psychological state of panic and fear. In addition, as illustrated in Figure 3, our generated motions exhibit not only diverse characteristics, such as the range of motion and which hands to use, but also a high degree of consistency with the ground truth.

## 4.6   Ablation Study

**Effect of VQVAEs.** We confirm the significant role of the VQVAEs. As shown in Table 2 and Table 3, the integration of VQVAEs is essential for the functionality of our approach, contributing to the generation of gestures that exhibit smoother transitions and a more human-like quality. As

demonstrated in Table 3, the removal of the VQVAEs ("−VQVAEs") from the model is also associated with performance decline, manifesting reduction in FGD, BC, Diversity, MSE and LVD.

**Effect of Local Scan.** We validate the effectiveness of the local scan. The ablation study is divided into two segments: (i) multi head cross attention and (ii) Mamba models for different part of bodys. As shown in Table 3, incorporating multi head cross attention enhances our method's capability to generate gestures with higher beat constancy. The incorporation of the Mamba from local scan generate gestures characterized by greater diversity. Concurrently, there is an observed improvement in the FGD of the generated gestures.

Table 3: Ablation study on different components of our proposed method. ↓ denotes the lower the better, and ↑ denotes the higher the better. FGD multiplied by $10^{-1}$, BC multiplied by $10^{-1}$, Diversity, MSE multiplied by $10^{-7}$, and LVD multiplied by $10^{-5}$.

| Method | FGD ↓ | BC ↑ | Diversity ↑ | MSE ↓ | LVD ↓ |
|---|---|---|---|---|---|
| Ours | 5.366 | 7.812 | 13.048 | 0.629 | 6.897 |
| − VQVAEs | 12.051 | 7.447 | 8.462 | 1.316 | 9.235 |
| − Local Scan ($\mathcal{F}_{MHCA}$) | 7.189 | 6.701 | 13.216 | 0.638 | 6.938 |
| − Local Scan (Mamba) | 7.277 | 7.742 | 12.844 | 0.627 | 6.941 |
| − Global Scan ($\mathcal{F}_{MHSA}$) | 6.308 | 7.882 | 11.875 | 0.644 | 6.972 |
| − Global Scan (Mamba) | 6.149 | 7.840 | 12.605 | 0.592 | 6.752 |

**Effect of Global Scan.** We validate the effectiveness of the global scan, as listed in Table 3, the incorporation of global scan improves the overall performance of our method. For the multi-head self-attention module in global scan, the incorporation of multi-head self-attention acquires improvement of Diversity and a degradation for FGD. Additionally, the ablation results demonstrate that incorporating Mamba enhances the global scan's capability to generate gestures with higher diversity. The FGD of generated gestures is better at the same time.

Table 4: Ablation study on different audio encoders. ↓ denotes the lower the better, and ↑ denotes the higher the better. FGD multiplied by $10^{-1}$, BC multiplied by $10^{-1}$, Diversity, MSE multiplied by $10^{-7}$, and LVD multiplied by $10^{-5}$.

| Method | FGD ↓ | BC ↑ | Diversity ↑ | MSE ↓ | LVD ↓ |
|---|---|---|---|---|---|
| Ours | 5.366 | 7.812 | 13.048 | 0.629 | 6.897 |
| Whisper [47] | 6.791 | 7.515 | 12.617 | 0.537 | 6.445 |
| Wav2vec2 [3] | 5.343 | 7.956 | 13.164 | 0.973 | 8.452 |

**Effect of Different Audio Encoder.** To validate the effectiveness of the audio encoder, we replace the CNN-based audio encoder with a pre-trained Whisper [47] and Vav2Vec2 [3], as listed in Table 4. Unlike CNN-based audio encoders that are randomly initialized and trained from scratch, when using Whisper and Wav2Vec2, we initialize the encoder using pre-trained weights and fix the parameters of the feature extractor. We observe a notable enhancement in facial generation when utilizing Whisper, however, the body generation results were subpar. In contrast, while Wav2Vec2 demonstrates some improvement in body generation, it results in a substantial decline in facial generation quality.

## 5  Conclusion

In this study, we propose a framework to employ the state space models in gesture synthesis. To alleviate the problem of jitter in gesture synthesis, we have implemented discrete motion priors, which enhance the effectiveness of the selective scan mechanism and lead to smoother results. We further incorporate the selective state space models with attention mechanisms to enhance the refinement of motion features in latent space. These modules capture the subtle movements and deformations of various body parts, thereby enhancing the overall quality of the generated gestures. By utilizing a linear time series modeling strategy with selective state space, our method achieves high-quality full body gesture generation with low latency.

## Acknowledgements

This work was supported by the STI 2030-Major Projects under Grant 2021ZD0201404 and in part by the National Natural Science Foundation of China under Grant 62306165.

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

# A  Appendix / Supplemental material

## A.1  More visualization results

Figure 4 presents the facial motion results generated by our method, showcasing the generation of facial expressions and movements with a high level of realism. Our method effectively synchronizes with the phonetic articulation of speech content, accurately reflecting the physical demands of pronunciation. For instance, when uttering "walking", "came" or "bus", our approach ensures that the mouth's movements, such as opening, correspond closely with the actual phonetic requirements. Other methods do not consistently achieve this level of accuracy in aligning with the phonetic and physical nuances of speech. Our method adeptly handles the subtleties of mouth closure and elongation required for sounds such as "in", closely aligning with the ground truth, whereas other approaches may exhibit inconsistencies in this regard. Moreover, in instances of silence, all methods, including ours, demonstrate a good capacity to learn and maintain the mouth's closed position, effectively reflecting the underlying patterns of speech and silence. Since CaMN does not specifically target the generation of facial movements, it results in a lack of variation in facial expressions throughout the process.

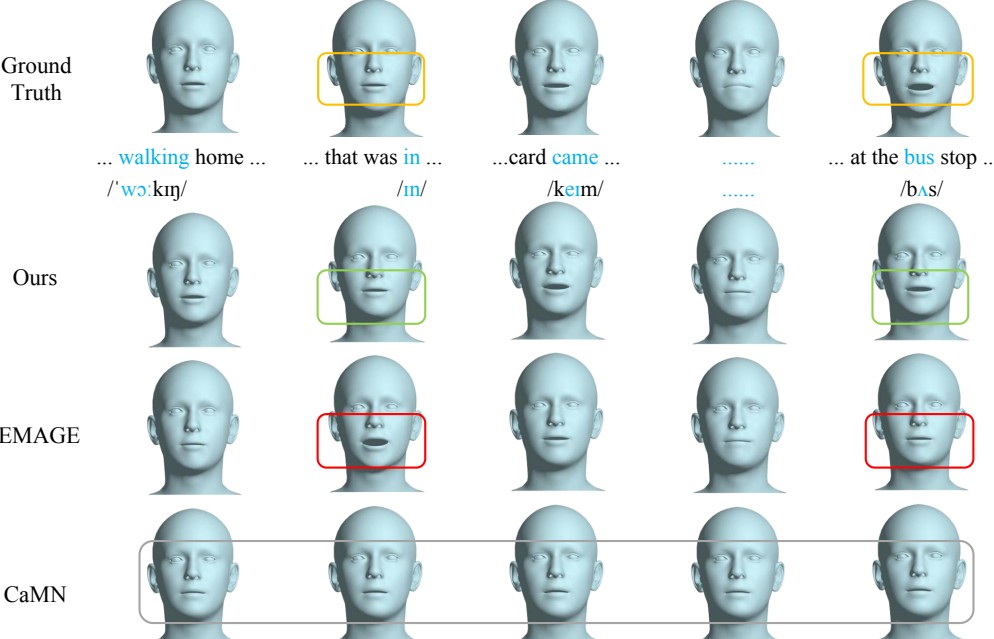

Figure 4: Visualization of the facial motions generated by CaMN, EMAGE and our method. Unreasonable results are indicated by red and gray boxes and reasonable ones by green boxes.

As illustrated in Figure 5, our methodology generates gestures that demonstrate improved rhythmic synchronization and a more lifelike appearance, effectively capturing the essence of the speaker's rhythmic patterns. For example, in the expression "actually" our method guides the individual to bring the hands inward in front of the chest, a subtle gesture not observed in the results produced by CaMN and EMAGE, where the arms are either hang limply at the sides or are splayed downward. Furthermore, in the depiction of "on the way back" our approach accurately reflects the ground truth by slightly bending down and raising one hand, while EMAGE cannot respond accurately to this and remains standing.

Furthermore, our approach accurately captures the semantic essence of movements. For instance, in response to the cue "hug" our method generates an inward-circling motion of the arms, aligning perfectly with the ground truth, which is a nuanced semantic element that other methodologies neglect. Similarly, in scenarios such as "so small", the result of our method is similar to the ground truth, with the character's hand moving inward. This attention to detail ensures semantic consistency, which is lacking in other approaches where actions are not aligned with the intended meaning.

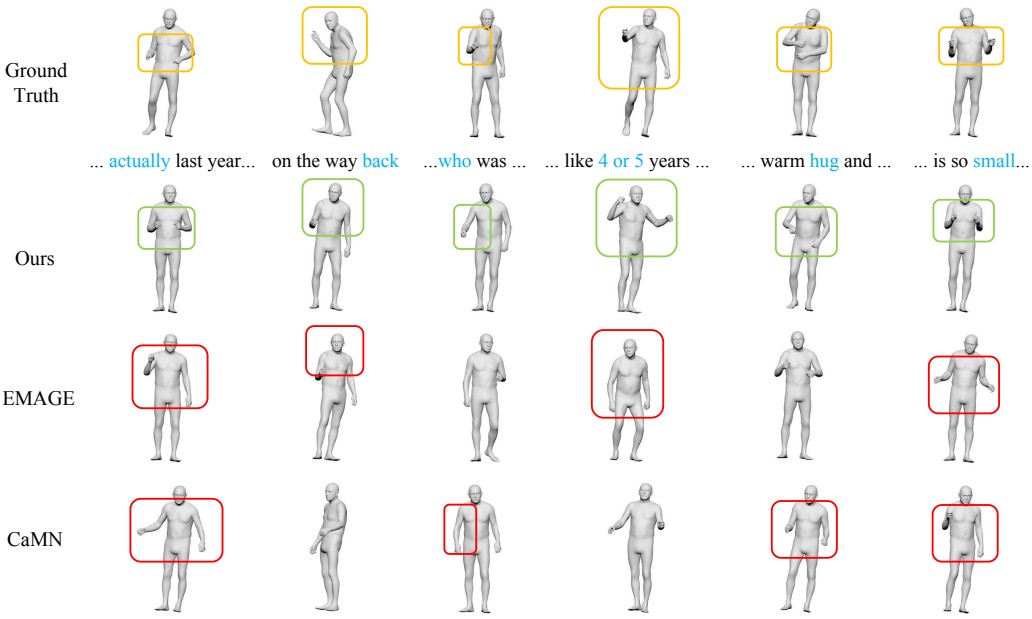

Figure 5: Visualization of the gestures generated by CaMN, EMAGE and our method. Unreasonable results are indicated by red boxes and reasonable ones by green boxes.

## A.2 Efficiency Analysis

We leverage the linear computational complexity of Mamba and the sequence compression capability of VQVAE within our framework, which helps in reducing computational complexity. Although there are some specialized acceleration solutions, faster solutions are necessary because when the model is integrated into the system, there is not only a delay in generating gestures, but also delays in other modules. Therefore, to evaluate the efficiency of our pipeline, we conducted a series of measurements, focusing on the runtime of individual components. We measured the runtime of various components on the NVIDIA A100 GPU in our method over three runs and presented the average results in Table 5. We also compare our method's inference time with diffusion-based methods. The average computation time was determined based on the generation of a 31 second motion sequence, to showcase our model's low latency capabilities. The total inference time of our method is much slower than state-of-the-art diffusion-based method [63]. The result confirm that our pipeline is well-suited for applications requiring low latency gesture generation like interactive systems, where responsiveness is paramount.

Table 5: The time cost for generating one second (average) of gestures using the method's modules.

| Modules | Run Time(s) |
| --- | --- |
| Diffusion-based method | |
| DiffStyleGesture [63] | $0.64365 \pm 0.0086$ |
| Our method | |
| Audio Encoders | $0.00217 \pm 0.0006$ |
| Text Encoders | $0.00480 \pm 0.0001$ |
| Global Scan | $0.00219 \pm 0.0004$ |
| Local Scan | $0.00676 \pm 0.0003$ |
| Face VQDecoder | $0.00073 \pm 0.0001$ |
| Hand VQDecoder | $0.00077 \pm 0.0001$ |
| Upper VQDecoder | $0.00106 \pm 0.0001$ |
| Lower VQDecoder | $0.00068 \pm 0.0001$ |
| Total Time | $0.01917 \pm 0.0018$ |

## A.3 Evaluation on BEAT dataset

To evaluate the generalisable benefit of our method, we conduct experiments on a large-scale multimodal dataset known as BEAT (Body-Expression-Audio-Text) [33]. This dataset encompasses 76 hours of multimodal data collected from 30 speakers engaging in conversations across four different languages while expressing eight distinct emotions. The dataset includes conversational gestures, facial expressions, emotional cues, and semantic content, along with annotations for audio, text, and speaker identity. To facilitate a fair comparison, we follow CaMN [33] and employ approximately 16 hours of speech data from English speakers. Furthermore, we implement the conventional approach of partitioning the dataset into distinct training, validation, and testing subsets, ensuring consistency with the data partitioning scheme utilized in prior research to uphold the integrity of the comparison.

To facilitate a fair comparison, we employ a total of N = 34 frame clips with a stride of 10 during the training process. The first four frames serve as seed poses, while the model is trained to generate the subsequent 30 poses, which collectively represent a duration of 2 seconds. Our models incorporate 47 joints from the BEAT dataset, comprising 38 hand joints and 9 body joints. As listed in Table 6, our method demonstrates a significant improvement compared to the CaMN (baseline), which also validates the generalisable benefits of our approach.

Table 6: Comparison with state-of-the-art method in the term of FGD, SRGR and BeatAlign. All methods are trained on BEAT datasets. ↓ denotes the lower the better while ↑ denotes the higher the better. The best results are in bold.

| Methods | FGD ↓ | SRGR ↑ | BeatAlign ↑ |
|---|---|---|---|
| Seq2Seq [71] | 261.3 | 0.173 | 0.729 |
| Speech2Gesture [11] | 256.7 | 0.092 | 0.751 |
| MultiContext [70] | 176.2 | 0.195 | 0.776 |
| Audio2Gesture [28] | 223.8 | 0.097 | 0.766 |
| CaMN [33] | 123.7 | 0.239 | 0.783 |
| TalkShow [68] | 91.00 | - | 0.840 |
| **MambaTalk (ours)** | **51.3** | **0.256** | **0.852** |

## A.4 Limitations

Currently, our approach to gesture synthesis involves using distinct modules to animate various body parts, which naturally introduces some latency. Developing a single, unified model capable of capturing the wide-ranging and intricate deformations and motion patterns characteristic of different body parts can be addressed in future research. This enhancement is anticipated to lower computational overhead and substantially reduce the processing time, thereby improving the real-time capabilities of the pipeline and ensuring a smoother and more responsive gesture generation system.

Meanwhile, exploring more robust audio representations or combining various types of pre-trained audio encoders could significantly enhance the quality of gesture generation. Our findings indicate that certain encoders, such as Whisper, are particularly effective for modeling facial movements, while others, like Wav2Vec2, are better suited for modeling body movements. This approach will further improve the overall performance of the method.

In addition, the issue of gesture diversity among speakers and across different cultures remains unaddressed. Addressing this gap is essential for improving the cross-cultural validity and expanding the applicability of gesture-based applications in diverse global contexts.

## A.5 Pseudo Code

The local scanning procedure is illustrated in Algorithm 25. We attain local modeling of various body segments by individually processing actions within distinct regions. The approach to global scanning parallels this methodology; however, the key distinction lies in the simultaneous processing of motion representations across multiple body parts.

---

**Algorithm 1** Local Scanning Process

---

**Require:** token sequence $\mathbf{T}_{l-1}$ : (B, M, D)
**Ensure:** token sequence $\mathbf{T}_l$ : (B, M, D)
  1: /* model motions in different body regions separately $\mathbf{T}'_{l-1}$ */
  2: $\mathbf{z_{face}}$ : (B, M, E) $\leftarrow$ $\mathbf{Linear^{face}}(\mathbf{T}'^{\,face}_{l-1})$
  3: $\mathbf{z_{upperbody}}$ : (B, M, E) $\leftarrow$ $\mathbf{Linear^{upperbody}}(\mathbf{SelfAttn}(\mathbf{T}'^{\,upperbody}_{l-1}))$
  4: $\mathbf{z_{lowerbody}}$ : (B, M, E) $\leftarrow$ $\mathbf{Linear^{lowerbody}}(\mathbf{SelfAttn}(\mathbf{T}'^{\,lowerbody}_{l-1}))$
  5: $\mathbf{z_{hand}}$ : (B, M, E) $\leftarrow$ $\mathbf{Linear^{hand}}(\mathbf{SelfAttn}(\mathbf{T}'^{\,hand}_{l-1}))$
  6: /* process with different parts of human body */
  7: **for** $o$ in {face, upperbody, lowerbody, hand} **do**
  8:      $\mathbf{x}'_o$ : (B, M, E) $\leftarrow$ $\mathbf{SiLU}(\mathbf{Conv1d}_o(\mathbf{x}))$
  9:      $\mathbf{B}_o$ : (B, M, N) $\leftarrow$ $\mathbf{Linear^B_o}(\mathbf{x}'_o)$
10:      $\mathbf{C}_o$ : (B, M, N) $\leftarrow$ $\mathbf{Linear^C_o}(\mathbf{x}'_o)$
11:      /* softplus ensures positive $\mathbf{\Delta}_o$ */
12:      $\mathbf{\Delta}_o$ : (B, M, E) $\leftarrow$ $\log(1 + \exp(\mathbf{Linear^\Delta_o}(\mathbf{x}'_o) + \mathbf{Parameter^\Delta_o}))$
13:      /* shape of $\mathbf{Parameter^A_o}$ is (E, N) */
14:      $\overline{\mathbf{A}}_o$ : (B, M, E, N) $\leftarrow$ $\mathbf{\Delta}_o \otimes \mathbf{Parameter^A_o}$
15:      $\overline{\mathbf{B}}_o$ : (B, M, E, N) $\leftarrow$ $\mathbf{\Delta}_o \otimes \mathbf{B}_o$
16:      $\mathbf{y}_o$ : (B, M, E) $\leftarrow$ $\mathbf{SSM}(\overline{\mathbf{A}}_o, \overline{\mathbf{B}}_o, \mathbf{C}_o)(\mathbf{x}'_o)$
17: **end for**
18: /* get gated $\mathbf{y}_o$ */
19: $\mathbf{y}'_{face}$ : (B, M, E) $\leftarrow$ $\mathbf{y}_{face} \odot \mathbf{SiLU}(\mathbf{z_{face}})$
20: $\mathbf{y}'_{upperbody}$ : (B, M, E) $\leftarrow$ $\mathbf{y}_{upperbody} \odot \mathbf{SiLU}(\mathbf{z_{upperbody}})$
21: $\mathbf{y}'_{lowerbody}$ : (B, M, E) $\leftarrow$ $\mathbf{y}_{lowerbody} \odot \mathbf{SiLU}(\mathbf{z_{lowerbody}})$
22: $\mathbf{y}'_{hand}$ : (B, M, E) $\leftarrow$ $\mathbf{y}_{hand} \odot \mathbf{SiLU}(\mathbf{z_{hand}})$
23: /* residual connection */
24: $\mathbf{T}_l$ : (B, M, D) $\leftarrow$ $\mathbf{Linear^T}(\mathbf{y}'_{face} + \mathbf{y}'_{upperbody} + \mathbf{y}'_{lowerbody} + \mathbf{y}'_{hand}) + \mathbf{T}_{l-1}$
25: Return: $\mathbf{T}_l$

---

### A.6 Evaluation Metrics

To evaluate the realism of body gestures, we employ Fréchet Gesture Distance (FGD)[70] to measure how close the distribution between the ground truth and generated body gestures is.

$$\mathrm{FGD}(\mathbf{g}, \hat{\mathbf{g}}) = \|\mu_r - \mu_g\|^2 + \mathrm{Tr}\left(\Sigma_r + \Sigma_g - 2\left(\Sigma_r \Sigma_g\right)^{1/2}\right), \tag{13}$$

where $\mu_r$ and $\Sigma_r$ denote the mean and covariance of the latent feature distribution $z_r$ for real human gestures $g$, while $\mu_g$ and $\Sigma_g$ correspond to the mean and covariance of the latent feature distribution $z_g$ for the synthesized gestures $\hat{g}$. We employ an encoder based on a Skeleton CNN (SKCNN) and a Full CNN-based decoder, constituting our autoencoder's pretrained network. This network is trained on both the BEATX-Standard and BEATX-Additional datasets. The preference for SKCNN over a Full CNN encoder stems from its superior performance in capturing gesture features, evidenced by a reduced reconstruction MSE loss of 0.095, as opposed to 0.103.

Subsequently, Diversity[28] is quantified by computing the average L1 distance across multiple body gesture clips. Higher Diversity signifies greater variance within the gesture clips. We compute the average L1 distance across various N motion clips using the following equation:

$$\text{Diversity} = \frac{1}{2N(N-1)} \sum_{t=1}^{N} \sum_{j=1}^{N} \left\| p_t^i - \hat{p}_t^j \right\|_1, \tag{14}$$

where $p_t$ denotes the positions of joints in frame $t$. We assess diversity across the entire test dataset. Moreover, when calculating joint positions, translation is zeroed, indicating that L1 Diversity is exclusively concentrated on local motion dynamics.

The synchronization between the speech and motion is conducted using Beat Constancy (BC)[30]. BC indicates a more precise synchronization between the rhythm of gestures and the audio's beat. We define the onset of speech as the audio's beat and identify the local minima of the upper body joints' velocity (excluding fingers) as the motion's beat. The synchronization between audio and

gesture is determined using the following equation:

$$\text{BC} = \frac{1}{g} \sum_{b_g \in g} \exp\left(-\frac{\min_{b_a \in a} \|b_g - b_a\|^2}{2\sigma^2}\right), \tag{15}$$

where $g$ and $a$ represent the sets of gesture beats and audio beats, respectively.

Turning focus to facial aspects, we gauge the positional accuracy through the calculation of vertex Mean Squared Error (MSE)[60]. This metric quantifies the average squared difference between the predicted facial landmarks and their corresponding ground truths, providing a clear indication of the facial model's accuracy:

$$\text{MSE} = \frac{1}{n} \sum_{i=1}^{n} (f_i - \hat{f}_i)^2, \tag{16}$$

where $n$ denotes the number of vertices, $f_i$ represents the ground truth position of the $i$-th vertex, $\hat{f}_i$ denotes the predicted position of the $i$-th vertex. The sum is taken over all vertices to compute the average error.

Additionally, the disparity between the ground truth and the generated facial vertices is measured by the vertex L1 difference (LVD)[68], which measures the synchronization between speech and facial expression.

$$\text{LVD} = \frac{1}{n} \sum_{i=1}^{n} \left\| f'_i - \hat{f}'_i \right\|_1, \tag{17}$$

where $n$ denotes the number of vertices, $f'_i$ represents the ground truth speed of the $i$-th vertex. $\hat{f}'_i$ denotes the speed of the $i$-th vertex in the generated facial expression. The sum is taken over all vertices to compute the average absolute difference.

## A.7 Ethical Considerations in Crowdsourcing Research

This section provides additional details about the user study for qualitative analysis. For our user study, we have randomly selected ten videos generated by different methods, each containing 20-second video clips. For each participant, we paid compensation that exceeded the local average hourly wage.

The screenshot of our user study website is illustrated in the Figure 6, which displays the template layout presented to the participants. In addition to the main trials, participants were also subjected to several catch trials. These trials involved displaying Ground Truth videos and videos with distorted motion. Participants who failed to score the GT videos higher and the distorted motion videos lower were considered unresponsive or inattentive and their data was not included in the final evaluation.

# Rate your score on these videos.

In this task you are presented with mutilple videos of animated virtual characters.

**All videos have sound, please listen to them!**

You will be asked to rate the videos based on four different criteria.

Please focus on gestures and the facial expressions of the characters.

Choose your preference score from left to right. The rating score range is 1-5, with 5 being the best.

Please press play in order to start the videos. You need to watch and listen videos at least once to be able to answer.

## Video 1

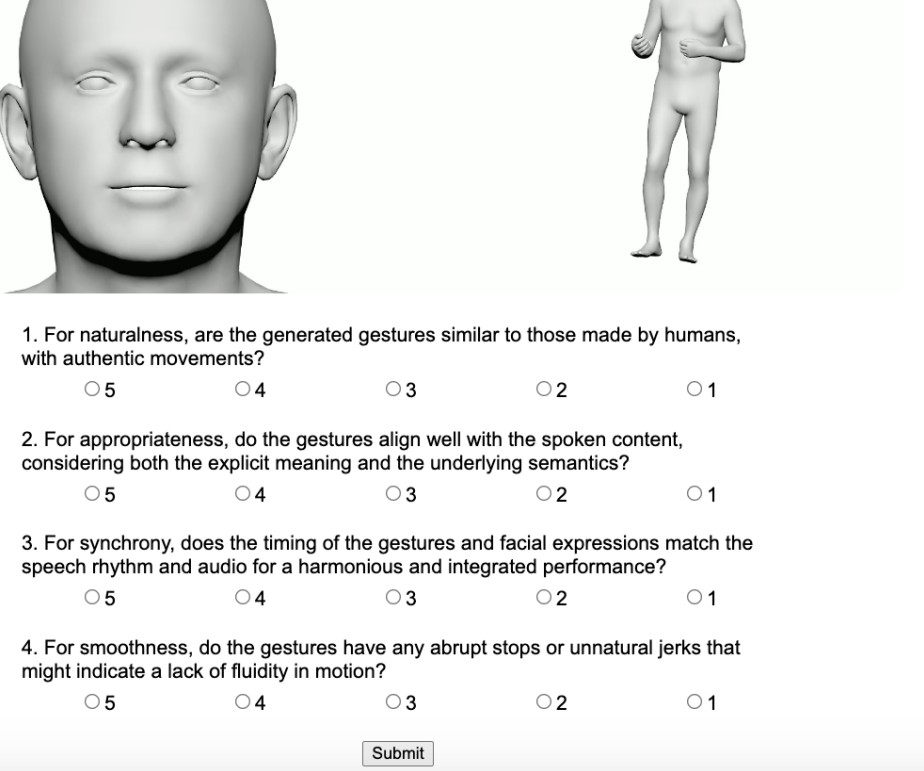

1. For naturalness, are the generated gestures similar to those made by humans, with authentic movements?

   ○ 5       ○ 4       ○ 3       ○ 2       ○ 1

2. For appropriateness, do the gestures align well with the spoken content, considering both the explicit meaning and the underlying semantics?

   ○ 5       ○ 4       ○ 3       ○ 2       ○ 1

3. For synchrony, does the timing of the gestures and facial expressions match the speech rhythm and audio for a harmonious and integrated performance?

   ○ 5       ○ 4       ○ 3       ○ 2       ○ 1

4. For smoothness, do the gestures have any abrupt stops or unnatural jerks that might indicate a lack of fluidity in motion?

   ○ 5       ○ 4       ○ 3       ○ 2       ○ 1

Submit

Figure 6: The screenshots of user study website for participants.

