# OpenReview forum: "MambaTalk: Efficient Holistic Gesture Synthesis with Selective State Space Models"
_NeurIPS.cc/2024/Conference — NeurIPS 2024 poster_

### Official Review · Reviewer_qFoz · 2024-07-04

**Soundness:** 4
**Presentation:** 4
**Contribution:** 3
**Rating:** 6
**Confidence:** 2

**Summary:**

This paper explores the application of state space models (SSMs) to co-speech gesture generation. The authors identify the computational challenges and jittering issues associated with the direct application of SSMs to gesture synthesis. To address these, they propose a two-stage modeling strategy with discrete motion priors and hybrid fusion modules. The first stage involves learning discrete holistic gesture priors with multiple VQVAEs, and the second stage refines latent space representations using local and global scans. Experiments demonstrate that MambaTalk outperforms state-of-the-art models in generating natural, rhythmic, and contextually appropriate gestures.

**Strengths:**

1. The paper is well-structured and clearly written.
2. It seems that MambaTalk is the first to explore the potential of the selective scan mechanism for co-speech gesture synthesis.
3. The methodology is thoroughly validated through extensive experiments, including both subjective and objective evaluations.

**Weaknesses:**

1. Limited novelty. Integrating the local and global scans seems not novel. More discussion on the comparison between MambaTalk and baseline methods is needed.
2. Limited datasets for method evaluation. Only the BEATX-standard dataset is adopted to evaluate the performance of the method. How about the zero-shot quantitative comparison on other datasets, such as the BEAT dataset or the RAVDESS dataset, which take emotion into account?

**Questions:**

See the "Weakness" section.

**Limitations:**

See the "Weaknesses" section. If possible, I suggest adding more visualization results and videos to the supplementary materials. Additionally, please note that the final score for this study is not solely determined by the peer reviewers' discussion. If the authors can address my main concerns, I would be willing to raise the score.

---

> ### Author Rebuttal · Authors · 2024-08-07
>
> **A1: Comparison with baseline methods**
>
> Thanks for your advice.
> Comparing with the baseline methods, we not only propose a method that applies a selective scan mechanism on co-speech gesture synthesis with local and global scans (a novel method not found in previous work), but we also consider the different motion patterns of different parts of the human body. We found that directly applying Mamba would cause serious shaking problems. Therefore, we leverage VQVAE and learnable queries to incorporate motion priors. Additionally, the direct application of Mamba also has the issue that the limb movements of different body parts tend to be average. Therefore, we integrate attention and selective scanning mechanisms into our framework to model spatial and temporal relationships. We will refine our statements and make our framework clearer (e.g., adding pseudo-code) in our revised version.
>
> Comparing to baselines, we have an advantage on all metrics, especially on BC (16.72%), MSE (36.35%), and LVD (13.99%). Our method also has efficiency in training and inference time (Appendix A.2). The training time for a single epoch of MambaTalk is only 42 seconds. In comparison, CaMN and EMAGE require 493 seconds and 83 seconds per epoch, respectively, which translates to 1173% and 212% of MambaTalk's epoch time. When considering the total time, MambaTalk also boasts a faster convergence rate, with our method requiring only 100 epochs.
> | Method | Time per Epoch (s) | Epochs |
> |--------------------------|------------------|-----------------|
> | CaMN     |  493(1173%) |  120 |
> | EMAGE   |    83(212%) |  400 |
> | Ours   |      42(100%) |  100 |
>
> **A2: Method evaluation**
>
> Thanks for your advice. Conducting a zero-shot comparison between the BEAT dataset, which uses BVH to represent gestures, and RAVDESS, represented by videos, would be challenging due to the use of SMPLX in BEATX.
> Therefore, we retrain our method using the BEAT dataset to consider emotions and verify the broad advantages of our approach for upper body movements. To incorporate emotions, we utilize embedding to convert them into style features and employ adaptive layer normalization to integrate them into the latent space of our framework.
> Due to time constraints, we only conducted preliminary experiments. The results are shown in the table. Our method demonstrates a significant improvement compared to the baseline, which also validates the effectiveness of our approach. We will conduct more comprehensive experiments and include additional details in the revised version.
> | Method | FGD$\downarrow$ | SRGR$\uparrow$ | BeatAlign$\uparrow$ |
> |--------------------------|------------------|-----------------|--------|
> | CaMN (baseline)     | 123.7   | 0.239       | 0.783   |
> | Ours   |       51.3        | 0.256             | 0.852     |
>
> **A3: More visualization results**
>
> Thanks for your kind advice. We will include more visualization results and videos in the supplementary materials in our revised version.

---

> > ### Comment · Reviewer_qFoz · 2024-08-10
> >
> > Thanks to the author for providing the rebuttal. The initial score remains the same after the rebuttal.

---

> > > ### Author Response · Authors · 2024-08-12
> > >
> > > We would like to express our sincere appreciation for your careful review and the time you have dedicated to evaluating our manuscript. We are grateful for the opportunity to provide a rebuttal and address the points you raised.
> > >
> > > We respect your decision and the thorough consideration you have given to our manuscript. We are pleased that our work has been recognized with a "weak accept" rating, which we interpret as a positive endorsement of our research.
> > >
> > > We would like to assure you that we have taken your feedback seriously and have made every effort to improve the manuscript. Should there be any further opportunities to refine our work or address any additional concerns, we are more than willing to do so.
> > >
> > > Once again, thank you for your valuable feedback.

---

### Official Review · Reviewer_p3w5 · 2024-07-09

**Soundness:** 2
**Presentation:** 2
**Contribution:** 2
**Rating:** 4
**Confidence:** 5

**Summary:**

This study explores the use of state space models (SSMs) to enhance gesture synthesis, addressing challenges such as diverse movement dynamics and unnatural jittering in generated gestures. Through a two-stage modeling approach and the introduction of MambaTalk with hybrid fusion modules, the study demonstrates superior performance compared to existing models in subjective and objective experiments.

**Strengths:**

1. This work is the first to explore the potential of the selective scan mechanism for co-speech gesture synthesis, achieving a diverse and realistic range of facial and gesture animations.
2. The writing of this work is fine, there are no obvious typos.
3. The workflow pipeline figure is clear to understand.

**Weaknesses:**

My concerns and suggestions of this manuscript are listed as blew:
1. The motivation of this work sounds unclear and looks a little incremental. As claimed in the abstract section 'the high computational complexity of these techniques limits the application in reality'. This statement is not clear. Does computational complexity mean more model parameters? or training time? or inference time? or more GPU memory cost? The authors didn't have a clear statement.
2. Moreover, in the abstract, 'which stem primarily from the diverse movement dynamics of various body parts.' are actually the general difficulties of the co-speech gesture generation task, not the specific one of how to employ the SSMs to this task.
3. The introduction section is not clear. The authors pay much more attention to the related works. Not the motivation and the high-level technical contributions of their work. This led me to feel that the entire work lacked technological innovation after reading the Introduction.
4. As for the methods, it is just directly imply the SSMs to this work. I cannot see any design on how to effectively solve the problem of 'computational complexity'. Actually, the technical contribution is poor, and the overall pipeline is very similar to the previous work EMAGE[1].
5. Could the authors explain why they only experimented on the BEATX dataset? As far as I know, TED[2] and TED-expressive[3] are also two commonly used co-speech gesture datasets.
6. The author did not conduct experiments to demonstrate how to effectively reduce the computational complexity by using SSMs, which directly led to the unclear motivation of this work.


[1] Liu, H., Zhu, Z., Becherini, G., Peng, Y., Su, M., Zhou, Y., ... & Black, M. J. (2023). Emage: Towards unified holistic co-speech gesture generation via masked audio gesture modeling. arXiv preprint arXiv:2401.00374.
[2] Yoon, Y., Cha, B., Lee, J. H., Jang, M., Lee, J., Kim, J., & Lee, G. (2020). Speech gesture generation from the trimodal context of text, audio, and speaker identity. ACM Transactions on Graphics (TOG), 39(6), 1-16.
[3] Liu, X., Wu, Q., Zhou, H., Xu, Y., Qian, R., Lin, X., ... & Zhou, B. (2022). Learning hierarchical cross-modal association for co-speech gesture generation. In Proceedings of the IEEE/CVF Conference on Computer Vision and Pattern Recognition (pp. 10462-10472).

**Questions:**

please refer to Weaknesses

**Limitations:**

please refer to Weaknesses

---

> ### Author Rebuttal · Authors · 2024-08-07
>
> **A1: Computational efficiency**
>
> Thanks for your question. Our computational efficiency is mainly reflected in the inference time, which has been analyzed in section A.2 of the appendix. We leverage the linear computational complexity of Mamba and the sequence compression capability of VQVAE within our framework, which helps in reducing computational complexity. Additionally, our method holds a significant advantage in training time. The training time for a single epoch of MambaTalk is only 42 seconds. In comparison, CaMN and EMAGE require 493 seconds and 83 seconds per epoch, respectively, which amounts to 1173% and 212% of MambaTalk's epoch time. When considering the total time, MambaTalk also demonstrates a faster convergence rate, as our method only requires 100 epochs.
> | Method | Time per Epoch (s) | Epochs |
> |--------------------------|------------------|-----------------|
> | CaMN     |  493(1173%) |  120 |
> | EMAGE   |    83(212%) |  400 |
> | Ours   |      42(100%) |  100 |
>
> **A2: Writing improvement**
>
> Thanks for your valuable comments. The goal of our work is also to use Mamba to address the difficulties of the co-speech gesture generation task. We will refine our statement in the abstract.
> For the introduction section, we will simplify the description of related work and move this part to the related work section to make space for our motivation and the high-level technical contributions of our work (as illustrated in A3: Comparison with EMAGE).
>
> **A3: Comparison with EMAGE**
>
> Our work does not simply involve implementing Mamba for gesture synthesis. We found that directly applying Mamba would cause serious shaking problems. Therefore, we incorporated motion priors using VQVAEs and individual learnable queries for different parts of the body. Our solution differs from EMAGE, which involves extracting motion cues from masked body joints.
> At the same time, the direct application of Mamba also has the issue that the limb movements of different body parts tending to be average. Therefore, we refine the design of spatial and temporal modeling in latent spaces by proposing a local-to-global modeling strategy and incorporating them with attention and selective scan mechanism into the design of our framework. Our work is also the first framework based on SSMs designed for co-speech gesture synthesis. We will refine our statements and make our framework clearer (e.g., adding pseudo-code) in our revised version.
>
> For comparison with the transformer-based method EMAGE[1], our SSM-based method also has an advantage over EMAGE in all metrics, especially in terms of BC (16.72%), MSE (36.35%), and LVD (13.99%) metrics.
> Our work is not a disruptive innovation. However, in addition to using SSMs instead of transformers, we have also developed numerous adaptive designs for SSMs to make it work.
>
> **A4: Reasons for choosing BEATX and the generalizable benefits of our method**
>
> Thanks for your question. We experiment on the BEATX datasets since our work focuses on holistic gesture synthesis.
> However, most current datasets only include movements of specific body parts, not the entire body.
> For example, the TED[2] and TED-expressive[3] datasets that you mentioned only focus on upper body gesture synthesis. TED includes 10 upper body joints, while TED-expressive includes 13 upper body joints and 30 finger joints.
> Moreover, TED and TED-expressive datasets are based on the 3D pose estimator ExPose for extracting gestures, which contain some errors in the ground truth.
> Therefore, we conducted experiments using another motion capture dataset called BEAT to validate the generalizable benefits of our method. This dataset contains data on upper body and hand movements to validate the generalizable benefits of our method for specific parts of body movements.
> Due to time constraints, we only conducted preliminary experiments. The results are shown in the table. Our method demonstrates a significant improvement compared to the baseline, which also validates the effectiveness of our approach. We will conduct more comprehensive experiments and include them in the revised version.
> | Method | FGD$\downarrow$ | SRGR$\uparrow$ | BeatAlign$\uparrow$ |
> |--------------------------|------------------|-----------------|--------|
> | CaMN (baseline)     | 123.7   | 0.239       | 0.783   |
> | Ours   |       51.3        | 0.256             | 0.852     |

---

> > ### Comment · Reviewer_p3w5 · 2024-08-11
> > **Response to authors**
> >
> > Dear authors,
> >
> > Thanks for your efforts and responses to my questions.
> > Although most of my concerns are addressed by the authors, I still have some main concerns about the motivation and writing of this work.
> > Therefore, I raise my rating from 3 to 4.
> > Thank you.
> >
> > Best regards

---

> > > ### Author Response · Authors · 2024-08-12
> > >
> > > We are grateful for your engagement with our rebuttal and for the opportunity to address your concerns. We understand that despite our efforts to respond to your initial questions, there are still aspects of our work that have not fully met your expectations. Your feedback is invaluable to us, and we are committed to enhancing the quality of our research.
> > >
> > > We acknowledge that our motivation and the clarity of our writing may not have been as compelling as they should be. To address these issues, we will take the following steps in our revised paper:
> > > - **Clarification of Motivation**: We will revisit the introduction and conclusion sections to more explicitly articulate the significance and novelty of our research. We have included additional context to underscore the importance of our work within the broader scope of co-speech gesture synthesis. For the motivation of long-term and real-time motion synthesis, this approach is critical for human-computer interactive scenarios, where real-time performance is crucial, and some conversations require extended periods of speech. In this context, we integrated VQVAE with Mamba in our framework, effectively addressing the shortcoming of diffusion models (e.g., slow inference speed) through VQVAE's compression expression ability and Mamba's linear computational complexity.
> > > - **Enhanced Writing Quality**: We will undertake a thorough review of the paper to ensure that the writing is clear, concise, and engaging. We will also seek the assistance of professional editors to refine our language and presentation.
> > > - **Addressing Specific Concerns**: We have carefully considered each of your points and provided targeted feedback. We believe these changes have strengthened the overall coherence and persuasiveness of our arguments. If you have any questions, please feel free to raise them.
> > >
> > > We hope that these revisions have adequately addressed your concerns and have brought our manuscript closer to acceptance. We are open to further suggestions and are willing to make additional revisions as needed to meet the standards of NeurIPS.
> > >
> > > Thank you once again for your constructive feedback.

---

### Official Review · Reviewer_WGJa · 2024-07-12

**Soundness:** 3
**Presentation:** 2
**Contribution:** 3
**Rating:** 6
**Confidence:** 3

**Summary:**

The paper explores the selective scan mechanism for gesture generation.  First, it trains VQVAE to reconstruct faces and body parts using discrete latent space. Then, It uses local and global scanning mechanisms to improve the latent representations of various body parts for the purpose of gesture generation.

**Strengths:**

1) The paper explores Mamba for gesture generation.
2) The objective function includes features related to gesture acceleration and velocity, which are important to capture.
3) The paper contains a user study where even the ground truths are evaluated. I think that is the right direction for gesture-related studies.

**Weaknesses:**

1) The related work contains many acronyms not introduced or known to a wide community (e.g., VQVAEs, HiPPO, and LSSL).
2) How the authors used Mamba in their work is unclear. The explanation in section 3.1 is vague.
3) Audio feature extractor is not strong. The authors should have explored stronger speech representations that can be used to improve speech prosody.
4) The technical contributions of the paper might be limited to implementing Mamba architecture for gesture analysis.

**Questions:**

1) Can you please dedicate section 3.1 to how you used Selective State Spaces for gesture generation? The explanation given is very generic. The reader can not follow this generic description without prior knowledge of Mamba.
2) How are the motion priors (the codebook) initialized? Also, how often is this codebook updated?
3) Why are the results reported in "EMAGE: Towards Unified Holistic Co-Speech Gesture Generation via
Expressive Masked Audio Gesture Modeling" differs from what you report in Table 1. FGD in the EMAG paper is 5.512, while in your paper, they are reported as 5.423. If you are using two different backbone models (and hence features), then how fair is this comparison if you fine-tune your models with the best parameters?

**Limitations:**

One limitation related to gesture diversity across speakers and cultures is not addressed. The mentioned limitations are mainly related to technical details, which I think are already being addressed using Transformer and Diffusion based models.

---

> ### Author Rebuttal · Authors · 2024-08-07
>
> **A1: Related work and Preliminaries section**
>
> We apologize for any inconvenience caused to your reading due to the generic explanation. We will provide a more detailed explanation for related work (e.g., VQVAE, HiPPO, LSSL) in section 3.1, which acts as preliminaries of our work.
> For the use of Mamba, we mainly discuss its application in Section 3.3, where we combine Mamba with attention mechanisms and learnable queries to model motion sequences using a local-to-global scanning strategy. In the revised version, we will supplement more details, including pseudo-code, to provide a clearer explanation of our method.
>
> **A2: Codebook**
>
> Thanks for your questions. We utilize uniform initialization for the codebook, with a numerical distribution range of [-1/codebook_size, 1/codebook_size), and conduct the first training stage on the training dataset. The codebook is solely updated during the first stage, and in the second stage of training for the speech-to-gesture mapping, the codebook remains frozen.
>
> **A3: Audio representation**
>
> Thanks for your valuable comments. Enhancing the audio representation can potentially improve performance, but it might also result in unfair comparisons with previous methods. In our revised version, we will include additional experiments using various audio feature extractors (e.g., wav2vec2, Whisper).
>
> **A4: Technical contribution**
>
> Our work did not simply involve implementing Mamba for gesture synthesis. We found that directly applying Mamba would cause serious shaking issues. Therefore, we explored incorporating motion priors using VQVAEs and individual learnable queries for different parts of the body.
> Additionally, the direct application of Mamba also has the issue that the limb movements of different body parts tend to be average. Therefore, we refined the design of spatial and temporal modeling in latent spaces by proposing a local-to-global modeling strategy and incorporating them with attention and selective scan mechanism into the design of our framework. We will refine our statements and make our framework clearer (e.g., adding pseudo-code) in our revised version.
>
> **A5: Results of EMAGE**
>
> For the difference in results, we compared the v3 version released by EMAGE on arXiv, which the authors of EMAGE claimed to be the CVPR 2024 camera-ready version.
> We checked again and found that they later updated multiple versions of arXiv. The latest version, v5, does indeed report 5.512. We will incorporate their updated results in our revised version.
> To ensure fairness in the comparison, we maintained consistency with them in the experimental setting. We have an advantage over EMAGE in all metrics, with significant performance improvements in BC, MSE, and LVD metrics, demonstrating enhancements of 16.72%, 36.35%, and 13.99%.
>
> **A6: Limitation section**
>
> Thanks for your valuable comments. We will add this discussion to the limitations section.

---

> > ### Comment · Reviewer_WGJa · 2024-08-13
> >
> > Thank you very much for the clarification and the response, which is much appreciated, especially the point related to the comparison of the result. I have raised my ratings from 5 to 6.

---

> > > ### Author Response · Authors · 2024-08-13
> > >
> > > We are deeply grateful for your thoughtful reconsideration of our manuscript and the increase in your rating (weak accept). Your feedback is instrumental in helping us refine our work, and we are pleased that our clarifications and responses have been well-received.
> > >
> > > Thank you once again for your constructive feedback and for the opportunity to improve our manuscript based on your valuable insights.

---

### Official Review · Reviewer_gANk · 2024-07-13

**Soundness:** 3
**Presentation:** 3
**Contribution:** 3
**Rating:** 5
**Confidence:** 5

**Summary:**

This paper focuses on the problem of co-speech gesture generation, particularly aiming to address the challenges of jittery movements, long motion sequences, and holistic gesture generation (including both face and body movements). To tackle these issues, the authors present a new method that combines diffusion models with state space models and incorporates both local and global scans. The proposed method has been evaluated on the BeatX dataset.

**Strengths:**

This paper introduces novel methodological aspects for co-speech feature generation. For instance, combining diffusion models with state space models and local and global scan approaches has not been explored before, to the best of my knowledge.

The experimental results clearly demonstrate the effectiveness of the proposed approach. Moreover, the paper presents both quantitative results and qualitative results through user studies, providing a complete and strong evaluation. Additionally, the proposed approach is faster compared to the state-of-the-art methods, as presented in the appendix.

However, there are some weaknesses that need to be addressed. Please see below for detailed comments.

**Weaknesses:**

Major Comment:

The main weakness of the paper is its limited evaluation. While the paper excels in generating facial, body, or holistic gestures, evaluating on only one dataset is substandard. To be competitive, the authors should consider additional standard benchmarks such as those used in the GENEA challenge [1]. There is also a long list of papers using diffusion models for co-speech gesture synthesis [1], none of which are mentioned in the paper. While I appreciate the authors' work and acknowledge the scarcity of datasets with both face and body motions, the proposed approach handles face and body separately. Therefore, at least body motion generation performance could be compared on existing datasets to demonstrate generalisable benefits.

Another weakness is the insufficient details regarding long-term motion generation, which is claimed as a key contribution. The paper lacks sufficient details on the length of the sequences considered in training and evaluation. Moreover, there is no comparison to support this claim.

Minor Comment:

The lower part of Figure 2 is helpful for understanding the workflow. However, including pseudo-code would be beneficial to increase the reproducibility of this method.

[1] https://genea-workshop.github.io/2023/challenge/

**Questions:**

Why is only one dataset considered? Please explain the rationale behind evaluating the method on only one dataset. How are these results generalisable and competitive given this limitation?

Generalizability and Competitiveness: How do you ensure that the results obtained from the BeatX dataset can be generalized to other scenarios? Discuss any potential limitations in terms of generalisability and how your method addresses them.

Training and Evaluation Sequence Lengths: Please elaborate on the length of the motion sequences considered during training and evaluation.

Impact on Co-Speech Gesture Generation Performance: Discuss how the length of motion sequences impacts the performance of co-speech gesture generation. What are the implications of using different sequence lengths, and how does your method handle long-term motion generation effectively? Including comparisons or experiments that highlight these aspects would be beneficial.

**Limitations:**

The authors have satisfactorily discussed the limitations and broader impact of their work.

---

> ### Author Rebuttal · Authors · 2024-08-07
>
> **A1: Reasons for choosing BEATX**
>
> Thanks for your kind advice. Our work focuses on holistic co-speech gesture generation. We chose to use this dataset because it includes global movements and the smplx sequences of the entire body (e.g., face, upper body, lower body, and hands). However, most current datasets (such as TED, TED-X, BEAT, GENEA) only include movements of specific parts of the body (e.g., upper body), not the entire body.
>
> **A2: Generalisable benefits of our method**
>
> To further validate the generalizable benefits of our method on upper body and hand movements, we used the BEAT dataset, which includes upper body and hand movements.
> Due to time constraints, we only conducted preliminary experiment. The results are shown in the table. Our method demonstrates a significant improvement compared to the CaMN (baseline), which also validates the effectiveness of our approach. More comprehensive experiments with detailed information will be included in the revised version.
> | Method | FGD$\downarrow$ | SRGR$\uparrow$ | BeatAlign$\uparrow$ |
> |--------------------------|------------------|-----------------|--------|
> | CaMN (baseline)     | 123.7   | 0.239       | 0.783   |
> | Ours   |       51.3        | 0.256             | 0.852     |
>
> **A3: Related diffusion-based work**
>
> Thanks for pointing out these wonderful works. For the diffusion-based model, we have compared our method with DiffuseStyleGesture in Table 1 and Table 4, which is a similar method to DiffuseStyleGesture+ (Reproducibility Award of GENEA workshop).
> Compare to DiffuseStyleGesture, DiffuseStyleGesture+ primarily considers the text modality as an additional input and utilizes channel concatenation to merge the text feature with the audio feature. Another leading solution (Diffusion-based co-speech gesture generation using joint text and audio representation) also incorporates the text modality as an additional input and employs contrastive learning to enhance the features.
> We will cite and further analyze the differences using all of these methods in our revised version.
>
> **A4: The length of the motion sequences considered during training and evaluation**
>
> Thanks for your professional advice. However, what we want to declare in our paper is that our method can generate long sequences with low latency (with analysis in Appendix A.2).
> For the length of the motion sequences considered during training and evaluation, we adopt a segmented modeling strategy. This strategy involves dividing the target sequence into multiple segments, each 64 frames in length, for processing. Therefore, our method can effectively handle long-term motion generation.
> The length of the motion sequence does not have a significant impact on performance because almost all segmented sequences share the same length and are processed in a similar manner.
>
> **A5: Pseudo-code**
>
> Thanks for your advice. We will include pseudo-code in our revised version. We will also make our code open-sourced if the paper get accepted.

---

> > ### Comment · Reviewer_gANk · 2024-08-11
> > **reply: Rebuttal by Authors**
> >
> > Thank you for the additional information. After reviewing all the feedback and responses, I have decided to maintain my initial score. While the paper introduces some novel methodological aspects, the primary concerns remain: the lack of experimental results and the need for further clarification, particularly regarding the motivation for long-term and real-time motion synthesis.

---

> > > ### Author Response · Authors · 2024-08-12
> > >
> > > We would like to extend our sincere gratitude for the time and effort you have invested in reviewing our rebuttal. We appreciate the detailed feedback and understand your concerns regarding the lack of experimental results and the need for further clarification on the motivation for long-term and real-time motion synthesis.
> > >
> > > For experimental results, we have conducted comprehensive experiments on holistic co-speech gesture synthesis (the main focus of our method), including quantitative results, qualitative results, and efficiency analysis in our initial version. In response to your feedback, we have incorporated supplementary experiment results on upper-body co-speech gesture synthesis in our rebuttal. The enhancement in our method is substantial, demonstrating improved overall performance compared to the baseline. Future experiments will focus on visualization.
> > >
> > > For the motivation of long-term and real-time motion synthesis, we expand our discussion on the motivation behind the need for long-term and real-time motion synthesis. This approach is critical for human-computer interactive scenarios, where real-time performance is crucial, and some conversations require extended periods of speech. In this context, we integrated VQVAE with Mamba in our framework, effectively addressing the shortcoming of diffusion models (e.g., slow inference speed) through VQVAE's compression capability and Mamba's linear computational complexity. As demonstrated in Appendix A.2, our method can generate sequences with extremely low latency, which is beneficial for the application of co-speech gestures.
> > >
> > > We look forward to your further feedback and are grateful for the opportunity to refine our work based on your valuable insights.
> > >
> > > Thank you once again for your consideration.

---

### Decision · Program_Chairs · 2024-09-25

**Decision:**

Accept (poster)

**Comment:**

The paper originally got BA, BR, R, and WA, and, after the rebuttal period, two reviewers raised the ratings, leading to final ratings of BA, WA, BR, and WA.

Reviewers recognized the major contribution of this paper in applying state space models (SSMs) into co-speech gesture generations. However, reviewers raised several concerns, including unclear motivation of the work in applying SSMs and insufficient experiments as the paper was tested on a single dataset. Following the rebuttal and discussion period, most concerns were addressed, and two reviewers increased their ratings, resulting in three positive and one negative review.

The AC also supports accepting the paper, recognizing its contribution to applying SSMs for the co-speech gesture generation. The AC strongly suggests the authors to follow the reviewers’ feedback in the camera-ready version.